# Effect of Finish Line Design on the Fit Accuracy of CAD/CAM Monolithic Polymer-Infiltrated Ceramic-Network Fixed Dental Prostheses: An In Vitro Study

**DOI:** 10.3390/polym13244311

**Published:** 2021-12-09

**Authors:** Mirza Rustum Baig, Aqdar A. Akbar, Munira Embaireeg

**Affiliations:** 1Department of Restorative Sciences (Prosthodontics), Faculty of Dentistry, Kuwait University, Kuwait 13110, Kuwait; 2Department of General Dental Practice, Faculty of Dentistry, Kuwait University, Kuwait 13110, Kuwait; aqdar.akbar@ku.edu.kw; 3Dental Department, Ministry of Health, Kuwait 13110, Kuwait; munira.embaireeg@gmail.com

**Keywords:** polymer/ceramic composites, biomaterials, marginal fit, internal gap, crown, computer-aided design and computer-aided manufacturing, resin, hybrid, urethane dimethacrylate, triethylene glycol dimethacrylate

## Abstract

A polymer-infiltrated ceramic network (PICN) material has recently been introduced for dental use and evidence is developing regarding the fit accuracy of such crowns with different preparation designs. The aim of this in vitro study was to evaluate the precision of fit of machined monolithic PICN single crowns in comparison to lithium disilicate crowns in terms of marginal gap, internal gap, and absolute marginal discrepancies. A secondary aim was to assess the effect of finish line configuration on the fit accuracy of crowns made from the two materials. Two master metal dies were used to create forty stone dies, with twenty each for the two finish lines, shoulder and chamfer. The stone dies were scanned to produce virtual models, on which ceramic crowns were designed and milled, with ten each for the four material–finish line combinations (n = 10). Marginal gaps and absolute marginal discrepancies were evaluated at six pre-determined margin locations, and the internal gap was measured at 60 designated points using a stereomicroscope-based digital image analysis system. The influence of the material and finish line on the marginal and internal adaptation of crowns was assessed by analyzing the data using two-way analysis of variance (ANOVA), non-parametric, and Bonferroni multiple comparison post-hoc tests (α = 0.05). ANOVA revealed that the differences in the marginal gaps and the absolute marginal discrepancies between the two materials were significant (*p* < 0.05), but that those the finish line effect and the interaction were not significant (*p* > 0.05). Using the Mann–Whitney U test, the differences in IG for ‘material’ and ‘finish line’ were not found to be significant (*p* > 0.05). In conclusion, the finish line configuration did not seem to affect the marginal and internal adaptation of PICN and lithium disilicate crowns. The marginal gap of PICN crowns was below the clinically acceptable threshold of 120 µm.

## 1. Introduction

Flexible resin matrix ceramic (RMC) materials have recently been introduced in dentistry for the computer-aided design/computer-aided manufacturing (CAD/CAM) fabrication of fixed indirect restorations, including single-tooth complete coverage crowns [1,2]. RMCs are purportedly able to overcome the shortcomings of traditional ceramics in that they can be applied directly after milling without the need for additional processing steps, such as firing, sintering, and glazing [2,3,4]. These materials are designed to combine the favorable mechanical properties of ceramic and resin into one single material, with flexural strengths and elastic moduli matching or being close to the natural tooth structure [4,5]. The polymer-infiltrated ceramic network (PICN) materials are a class of RMCs made up of a porous ceramic scaffolds infused with a mixture of urethane dimethacrylate (UDMA) and triethylene glycol dimethacrylate (TEGDMA) polymers [2,3,4,5]. Several recent studies have investigated different aspects of the PICN material related to its clinical application in restorative dentistry [3,5,6,7,8,9].

Marginal and internal adaptation are important criteria used to determine the clinical acceptability of fixed restorations at placement and success at future evaluations [10]. Although there are no clear-cut quantitative guidelines on the maximum allowable gaps at the tooth–restoration junction, a range of 80–120 µm is generally considered as clinically acceptable for ceramic CAD/CAM single crowns, according to systematic reviews [11,12] and other relevant guidelines [13]. In several studies, multiple factors have been investigated for their possible effect on the fit accuracy of ceramic crowns, including the ceramic material characteristics and processing method, the CAD/CAM system type, the measurement technique employed, and the preparation design variations [11,12,14,15,16]. However, conclusive evidence on the effect of these variables on the precision of fit is still unavailable.

A number of recent studies [5,6,17] have investigated the microstructural and mechanical properties of contemporary monolithic dental restorative materials, including PICN hybrid ceramic. Recent review papers [2,3] have also highlighted the favorable mechanical properties and bond strength of PICN, as well as the potential for the clinical use of indirect fixed restorations, including crowns. However, information on the marginal and internal adaptation of PICN crowns is lacking, especially in relation to different preparation designs [7,8,9].

Most studies have assessed the vertical gap and internal fit of ceramic crowns [8,11,12,14,16,18], but the AMD, which also includes the horizontal discrepancy (overhang) component, has generally not been examined. The restoration of overcontour or undercontour (overhang or step) may have a negative effect on the periodontal tissues and may even lead to secondary caries due to plaque accumulation, and thus needs to be identified [19,20]. The finish line geometry has been shown in some studies to affect the marginal and internal fit of CAD/CAM ceramic crowns [21,22], but other papers have also found no significant differences related to this variable [16,23], with one study even showing mixed results [18]. A contemporary systematic review has highlighted the link between the finish line configuration and the marginal fit of ceramic crowns [14], but clear evidence on the role of this factor is unavailable in the literature, particularly with newer hybrid ceramic materials.

The objective of this laboratory study was to assess the fit accuracy of complete coverage monolithic PICN crowns in relation to LDS crowns in terms of MG, IG, and AMD between the crowns and the abutment (conforming to a maxillary premolar tooth). Secondly, the effect of the finish line preparation design on the precision of fit was also evaluated for the two materials. The null hypothesis was that there would be no differences in the fit, in terms of MG, IG, and AMD, between the PICN and LDS crowns. The second null hypothesis was that there would be no differences in the fit parameter values between the two margin types, shoulder and chamfer, for both the materials.

## 2. Materials and Methods

### 2.1. Preparation of the Master and Working Dies

Cast metal dies (Remanium 800 Cobalt Chromium alloy; Dentaurum, Ispringen, Germany) derived from ivorine maxillary first premolar tooth preparations (Columbia Dentoform Corp, Long Island City, NY, USA) were used as master models in this investigation (Figure 1). The two dies were provided with a 1 mm-wide continuous, internally rounded shoulder or a 1 mm-wide rounded chamfer, attained using a high-speed handpiece (KaVo Bella Torque Mini; KaVo, Lake Zurich, IL, USA) and burs (847 KR-016 KR taper modified shoulder, 850 KR-016 round end (chamfer), NTI Diamond Instruments; Kahla–GmbH, Thuringia, Germany). A 20° overall taper and 4 mm-axial cervico-occlusal height were maintained on the dies [16,18,24].

Forty polyvinyl siloxane impressions (light-body Aquasil ultra and heavy-body Aquasil; Dentsply De Trey, Konstanz, Germany) were made out of the two master metal dies to generate 40 type IV dental stone dies (20 chamfer and 20 shoulder). The dies were left to set for at least 24 h and were checked visually and by microscope (BM-1 stereo-microscope at 10×; Meiji Techno, Saitama, Japan) for any nodules or voids by one of the authors. Once found to be satisfactory, the chamfer and shoulder samples were each divided into 2 groups of 10 dies randomly by numbering the dies and drawing lots; these were then allocated to the two crown systems: PICN (Enamic, Vita Zahnfabrik GmbH, Bad Säckingen, Germany) and LDS (IPS e.max CAD, Ivoclar Vivadent AG, Schaan, Liechtenstein).

### 2.2. Preparation of the Crown Specimens

Laboratory scans of the stone dies were performed using a digital scanner (Medit T710, Seoul, Korea), producing 10 shoulder and 10 chamfer scans for each material. Complete contour wax-ups were carried out on two of the stone die samples (one chamfer and one shoulder) to be used as reference templates for the manufacture of standardized monolithic crowns (PICN and LDS). The stone dies were digitized with the wax-up using the laboratory scanner (Medit T710, Seoul, Korea). The wax-up scans were then superimposed on the stone die scans to design virtual crowns which conformed to the same shape and contour for all the crown samples for each material and finish line, using the CAD software (Dental CAD 3.0, Exocad GmbH; Darmstandt, Germany) with a 0.03 mm cement space setting. A five-axis milling machine (CEREC InLab MC X5, Dentsply Sirona, Charlotte, NC, USA) was used to cut the monolithic ceramic crowns from IPS E.max CAD (IPS E.max CAD LT A1/C14, Ivoclar Vivadent AG Schaan, Liechtenstein) and PICN Enamic Blocks (Vita Zahnfabrik GmbH, Bad Sackingen, Germany). Specific milling bits and grinding pins were used to machine the two materials. Fresh sets of milling bits were used for each of the four material–finish line groups, thus yielding 10 crowns per set.

The milled LDS crowns were crystallized using a ceramic furnace (Programat P310; Ivoclar Vivadent AG, Schaan, Liechtenstein) at 850 °C for 25 min, as per the manufacturer’s recommendation. The PICN Enamic crowns did not undergo any additional procedures after milling. The fit of all the crowns was assessed on the individual stone dies using a microscope (X10) and they were finished and polished (Porcelain adjustment kit HP and Porcelain veneer kit HP, Shofu finishing and polishing systems; Shofu Inc., Kyoto, Japan). Finally, 40 crowns were prepared for marginal and internal fit evaluation (10 PICN-S, 10 PICN-C, 10 LDS-S, 10 LDS-C). The sample size of this study was calculated based on previous similar papers [7,8,9] and was estimated at 10 crowns for each of the two ‘ceramic material’ groups, based on the mean differences and standard deviation assumptions, at α = 0.05 and a power of 80% (G* Power statistical power software v.3.1.9.7, Kühbach, Germany) for the MG and IG evaluation. For the cervical finish line type, the sample size was calculated at 10 samples to elicit differences, achieving a power of 85% at α = 0.05.

### 2.3. Measurement of Marginal and Internal Fit

Six locations were selected for MG and AMD evaluation on each die: mesio-lingual line angle, ML; mid–lingual, MidL; disto-lingual line angle, DL; disto–buccal line angle, DB; mid–buccal, MidB; and mesio-buccal line angle, MB (Figure 2) [16,18,24,25,26]. The six sites were indented on the master metal dies with a round diamond bur (801-018 NTI Diamond Instruments; Kahla GmbH, Thuringia, Germany) around 2 mm below the preparation cavosurface margin. The dies were then partially embedded and retained in brass cylinders with acrylic resin (GC Pattern resin; GC Corp, Tokyo, Japan).

MG and AMD of the crowns were assessed directly on the master metal dies using a custom spring-loaded measuring jig. The brass cylinder with the crown seated on the master die was held in the jig under an occlusal load of 1.3 kg, delivered through a spring-loaded piston attached to a digital pressure gauge (MG20, Mark-10 Corp, Copiague, NY, USA) (Figure 3) [16,18,24]. Set screws enabled the rotation of the crown–die assembly around an axis passing through the center of the brass cylinder, allowing the observation of the crown–tooth junction at any required position.

The crown margins were examined and measurements were carried out at the six sites using a computerized digital image analysis system comprising a stereomicroscope (SZX7-ILST-SET stereomicroscope; Olympus Corp, Tokyo, Japan), a camera (X-cam, The Imaging Source Asia Co Ltd., Taiwan, PRC), and an image processing software (Cell D Image Analysis Software Ver. 3.1, Olympus Corp, Tokyo, Japan). Live images recorded using the stereomicroscope and camera set-up were streamed onto the computer monitor screen for examination. The processed images were confirmed for clarity, positioning, and lighting, and once they were found to be suitable they were saved for measuring MG and AMD. The stereomicroscopic measurement system was calibrated using an etched glass reticle measuring 1 mm. The reticle was focused under the microscope to enable horizontal and vertical linear measurements on the digitized images displayed on the computer monitor screen in microns. This was repeated multiple times to calculate the lengths, and the accuracy of the system was calculated to be in the range of ±5 µm.

The MG and AMD were measured according to the method described in previous studies [15,18,27], based on Holmes et al. [28], as illustrated in Figure 4.

All the crowns were seated and stabilized on the respective shoulder and chamfer master metal dies for performing MG and AMD measurements. The measurements were carried out at standardized sites using the dimples on the dies. The stereomicroscope was focused at the crown–abutment junction, with the concave part of the indentation meniscus showing clearly in each image. The process ensured that the same six sites were examined for each crown placed on the master metal die. A representative saved stereomicroscopic image of the PICN shoulder crown–tooth junction, as seen on the computer monitor at the mid-buccal marginal evaluation location, is shown in Figure 5.

A silicone replica technique was used to measure the IG. The master die was fitted with the individual crowns by injecting light-body silicone (Aquasil Ultra XLV Regular Set, Dentsply Detrey GmbH, Konstanz, Germany) into the intaglio surface of the ceramic crowns by applying finger pressure and holding it until the silicone material set. The procedure was performed by a single operator. This thin layer of silicone film was then fixed using heavy-body silicone (Aquasil Ultra Rigid Regular Set, Dentsply Detrey GmbH, Konstanz, Germany). The obtained silicone replica mold was cross-sectioned using a surgical scalpel blade, dividing it (No. 15) into six zones—buccal, mesio-buccal, mesio-lingual, lingual, disto-lingual and disto-buccal,—as shown in Figure 6.

In total, 60 pre-designated points were measured for IG, with 10 on each cut section (5 axial and 5 occlusal), as demonstrated in Figure 7. The points for the axial wall IG were chosen to be equidistant from each other (approximately 0.7 mm apart) starting from the inner curvature of the cervical finish line. Similarly, for the occlusal wall IG, five readings were recorded with approximately equal spacing between the points on the silicone layer. All measurements were performed using the stereomicroscope system used for MG and AMD determination following a similar technique.

The MG, AMD, and IG measurements were all carried out by the same assessor (M.E), at each of the 6 margin locations [29]. An average of 3 measurement repetitions at every marginal site were recorded as the final value. Intra-operator reliability assessment showed intra-class correlation coefficients of 0.95 for MG and 0.90 for AMD (α = 0.05), suggesting excellent agreement for the investigator for the measurements at different points. Additionally, a second assessor (M.R.B) randomly selected and performed the measurements on some samples, for marginal fit, to check the accuracy of the recorded readings. A high inter-operator reliability was confirmed through intra-class correlation coefficients of 0.91 for MG and 0.88 for AMD (α = 0.05).

### 2.4. Statistical Analysis

The mean values from all the measurement locations for each abutment tooth were calculated and the data were statistically analyzed (Statistical software SPSS v. 25; SPSS Inc., Chicago, IL, USA). The normality of the AMD, MG, and IG data was ascertained using Kolmogorov–Smirnov and Shapiro–Wilk tests. Based on the results, a two-way analysis of variance (ANOVA) was used to determine the effects of ‘material’ and ‘finish line’ on MG and AMD (α = 0.05). The non-parametric Mann–Whitney U test was employed to detect the influence of ‘material’ and ‘finish line’ on the IG (α = 0.05). The data were further analyzed using Bonferroni multiple comparison post-hoc tests to detect the differences between the various material–finish line groups regarding the MG, AMD, and IG parameters (α = 0.05).

## 3. Results

Table 1 shows the overall mean MG and AMD values for all the six marginal fit measurement sites combined for the two margin types, PICN and LDS crowns.

## 4. Discussion

Considering the significant differences found in the MG and AMD values between the two material groups, the first null hypothesis was partially rejected. However, in terms of IG comparison for the two materials, the results affirm this part of the first null hypothesis. With regard to the effect of margin configuration, in terms of MG, AMD, and IG, since there were no differences between the shoulder and chamfer designs, the second null hypothesis failed to be rejected.

The mean ± sd MG values obtained in this study for PICN crowns (36 ± 24 µm) were close to the numbers (51 ± 15 µm) reported in a recent PICN crown fit study [9], and also matched the lower side of the MG range (38–81 µm) seen in another similar study [8]. With regard to the AMD, two studies found values of 147 µm [9] and 183–212 µm [7] with PICN crowns, much higher than the current mean AMD value of 109 µm. So, generally, the marginal discrepancies in this study were found to be lower than those reported for PICN crowns in the literature for both MG and AMD parameters. The possible factors for the differences between the present data and previous results could be related to the CAD/CAM system used, the luting space settings, the measurement technique employed, the number of fit evaluation sites, cementation, and technical variations in the adjustment and finishing of the crowns. The laboratory scanner (Medit T710, Seoul, South Korea), the milling machine (CEREC InLab MC X5, Dentsply Sirona, Charlotte, NC, USA), and the design software (Dental CAD 3.0, Exocad GmbH; Darmstandt, Germany) used in this study are relatively new to the market and thus there are very few available fit accuracy studies peformed using this CAD/CAM system and this specific combination, especially in relation to PICN and LDS CAD monolithic crowns. Hence, the results of this study could not be compared further with reports of an equivalent kind.

The mean MGs of monolithic LDS crowns in this report (57 *±* 23 µm) corresponded with the outcomes of most other studies which investigated LDS CAD crown marginal fit [18,30,31,32]. However, there were some studies which reported numbers of 25–30 µm on either side of the general range seen (50–80 µm) [33,34]. The potential reasons that can be attributed to the dissimilarities are the crystallization parameter differences, abutment tooth type, the preparation design, and procedural variations in the labroatory processing of the crowns.

The mean MGs in this report for PICN and LDS crowns were below the reference value of 120 µm, commonly considered as a clinically acceptable threshold for ceramic crowns [13]. The MGs were also below the limit of 65–80 µm shown for CAD/CAM ceramic crowns in a contemporary review [12]. Although statistically significant differences were found in MG between the PICN (36 µm) and LDS (57 µm) crowns in this investigation, the magnitude of the disparity was small and the clinical significance of such a difference is unknown. The mean AMD values were markedly higher than the MG values in this paper for both the materials, in line with the findings of previous studies [7,9,15].

The current mean internal gaps in this study for all the material–finish line groups were in the range of 143–187 µm. These findings concurred with current reports on PICN crowns, which documented AMD values of 171–203 µm [7] and 150 µm [9]. The values also reconciled with the mean IG range provided in a systematic review [12] for CAD/CAM crowns (105–383 µm). However, the present IGs were higher than the 100 µm proposed by Molin et al. [35], as the maximum internal gap allowable for optimal clinical results with adhesive resin luting agents. Nevertheless, based on the IG values reported in the literature, the current numbers seem acceptable.

In this report, the margin type did not exert any significant influence on the fit accuracy of PICN and LDS ceramic crowns. The results are in agreement with those of several other studies which reported no significant differences between the two margin configurations, shoulder and chamfer [16,23,29]. These findings also concurred with the manufacturer’s recommendation of preparing either shoulder or chamfer margins for IPS e.max CAD crowns. However, the outcomes also differed from a few studies which showed that shoulder margins produced better MGs and AMDs compared to chamfer in ceramic crowns [18,21,22]. Additionally, the current study disagreed with the latest systematic review which showed significantly smaller marginal gaps with shoulder margin preparations compared to chamfers for ceramic crowns [14]. The reasons for this disparity could be caused by the basic differences in the fabrication modes of the two materials. Although a single technician produced the crowns using the same CAD/CAM system, the milling burs used in the exercise varied. Secondly, the crown materials were inherently distinct in their properties, with LDS undergoing an additional crystallization firing step, which the PICN did not.

There are some possible limitations in this paper that warrant discussion. The crowns were not cemented on the dies in this study prior to the fit evaluation as the same master metal dies were used for the entire process of fit accuracy determination. This could be considered a limitation because the clinical scenario might not be entirely simulated without cementation. The metal dies were generally non-abradable and dimensionally stable during the assessment procedure, permitting crown fit measurement to be carried out in a non-destructive manner. Scanning electron microscopy would have required cementation and the cutting of the crown–abutment complex to enable fit evaluation in addition to providing a restricted number of sections to perform the measurements. The micro-computed tomography technique would have been an appropriate choice for fit measurement, but the stabilization of the uncemented crown on the abutment during the scan procedure, using adequate pressure, would have been difficult. In addition, access to this sophisticated equipment is required for the research. Despite the many advantages of the stereomicroscopic technique, it can only perform a two-dimensional evaluation by direct measurement, unlike other three-dimensional techniques, and this could be deemed a small limitation.

Another slight limitation might be related to the abutment preparation design with a flat occlusal surface and margin at the same level horizontally. Even though many studies have used a similar abutment design for fit evaluation purposes, the design may not have accurately replicated the clinical situation. It must also be mentioned that gap measurement was not possible by direct viewing using the stereomicroscope in the proximal areas of the abutment (master die) because of the narrowing of the preparation in the mesial and distal regions of the tooth. Absolute values of AMDs were used in this study for data analysis. The AMDs generally represent the aggregate of vertical and horizontal marginal discrepancies at any given location at the crown–abutment junction [15]. However, the AMD values do not reflect the negative or positive overhang at an examined site, and this could be considered a minor limitation of the study.

Studies comparing PICN crowns fabricated using different preparation designs and fabrication techniques may be required to further advance the outcomes reported in this paper. Additionally, pitting the PICN restorations against the newer, fully crystallized versions of machinable hybrid glass-ceramics (zirconia lithium silicate) would be worthwhile. The results of this study should be interpreted with caution, as the accuracy of the clinical fit of restorations may not be the same as that achieved in a controlled laboratory environment. The clinical evaluation of the fit accuracy of PICN material with the new CAD/CAM systems and intra-oral scanners will aid in validating the results of this in vitro study.

## 5. Conclusions

Despite the limitations of the method employed in this study, it can be concluded that:There were significant differences in the MG and AMD of PICN crowns compared to LDS crowns (*p* < 0.05).For both PICN and LDS, no significant differences were found between the two finish line designs for all the three fit parameters examined (*p* > 0.05).The mean marginal gaps of the PICN and LDS crowns were below the proposed clinical acceptable limit of 120 µm for ceramic crowns and within the range reported in related review papers [11,12].

## Figures and Tables

**Figure 1 polymers-13-04311-f001:**
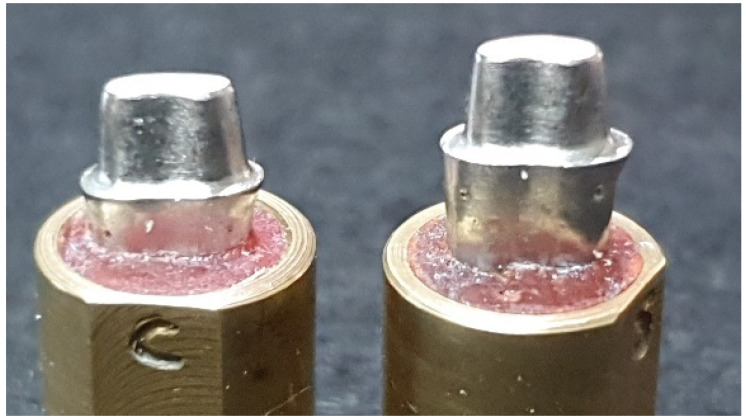
Master metal dies: chamfer and shoulder (left to right).

**Figure 2 polymers-13-04311-f002:**
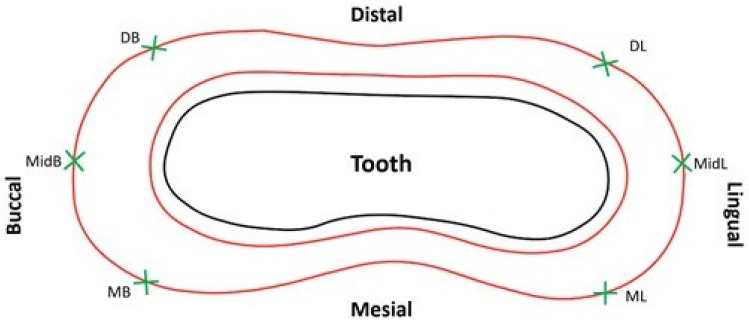
Six marginal fit evaluation sites around the periphery of the tooth.

**Figure 3 polymers-13-04311-f003:**
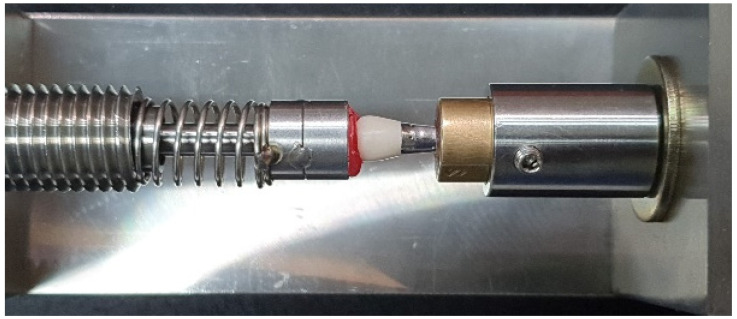
Marginal fit evaluation device with a crown fitted on the metal die.

**Figure 4 polymers-13-04311-f004:**
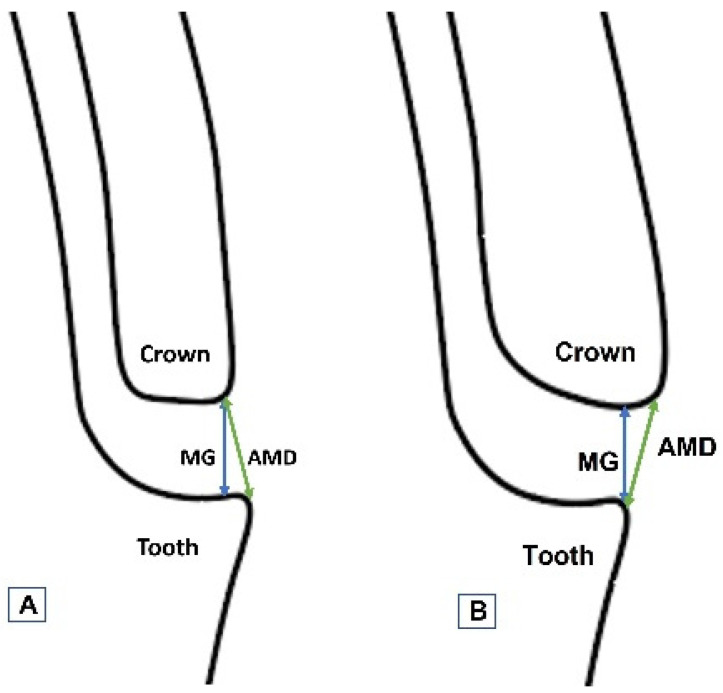
Schematic diagram showing the MG and AMD measurement scheme at the crown-abutment junction for undercontoured (**A**) and overcontoured (**B**) crowns.

**Figure 5 polymers-13-04311-f005:**
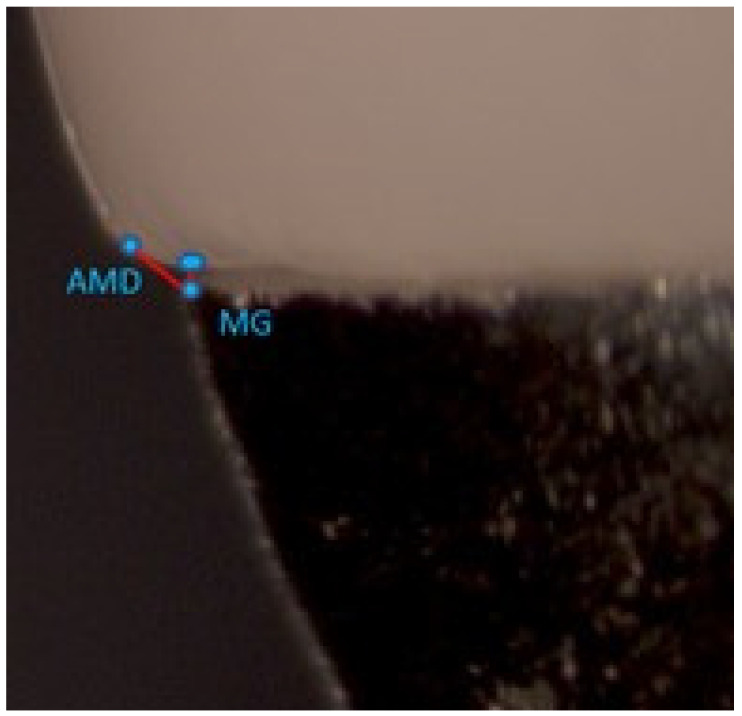
Representative image of the crown–abutment marginal junction showing MG and AMD measurements.

**Figure 6 polymers-13-04311-f006:**
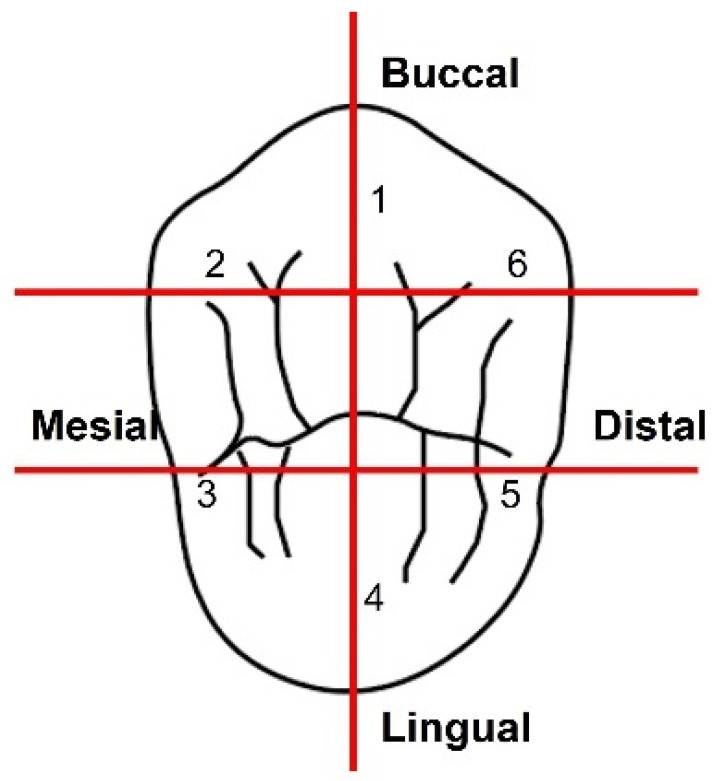
Schematic diagram of the occlusal view of the silicone replica mold showing the pattern of cut sections for IG evaluation: 1.buccal; 2. mesio-buccal; 3. mesio-lingual; 4. lingual; 5. disto-lingual; 6. disto-buccal(indicated through the red lines).

**Figure 7 polymers-13-04311-f007:**
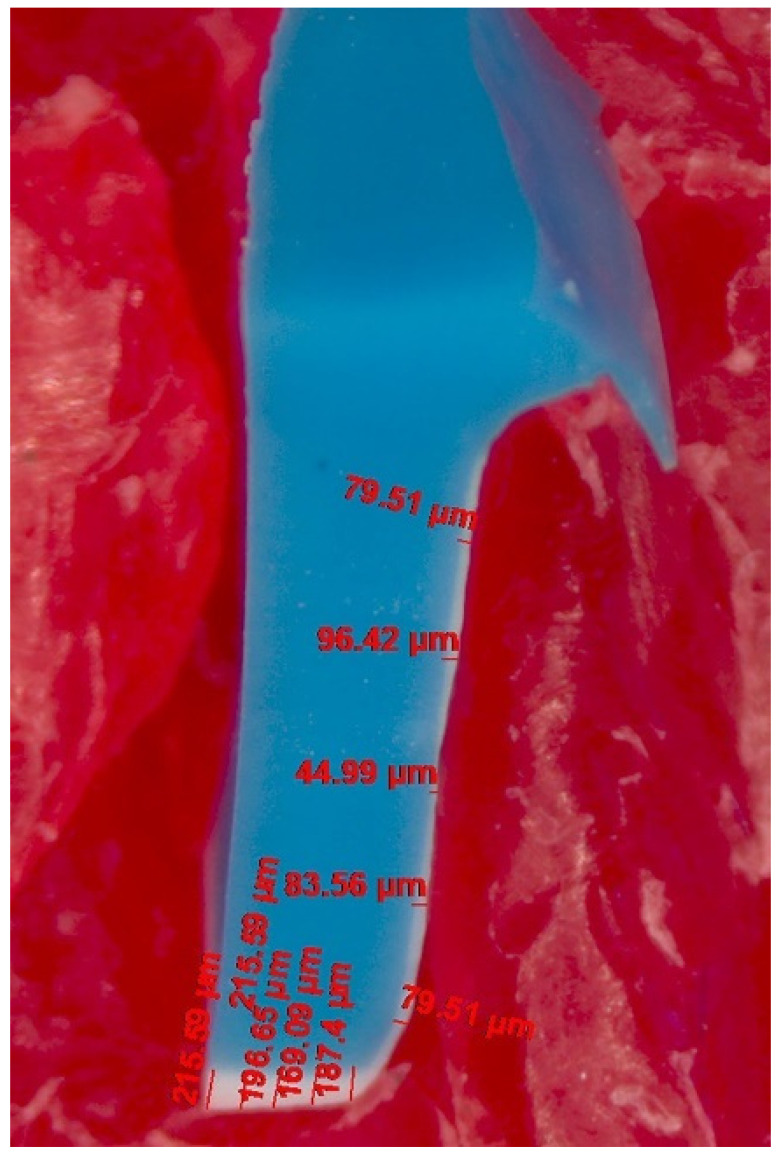
Stereomicroscopic image of the silicone replica cut section (indicated by the white layer) for PICN crown (zone 2) showing the IG measurement scheme on the axial and occlusal wall with representative values.

**Figure 8 polymers-13-04311-f008:**
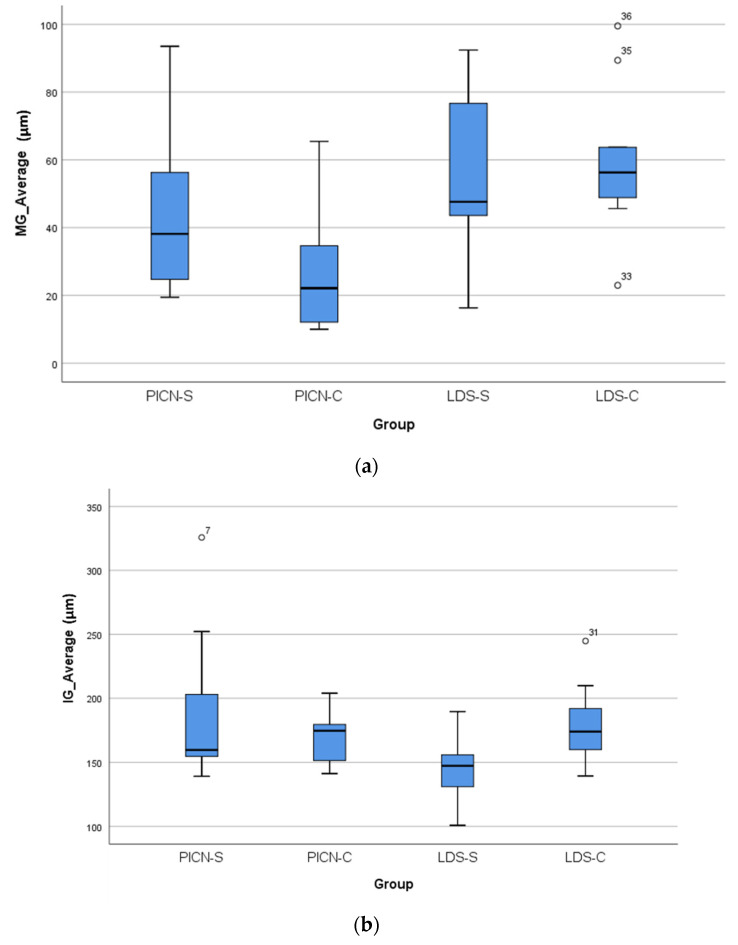
Box plot graphs showing the distribution of minimum, first quartile, median, third quartile, and maximum MG (**a**), IG (**b**), and AMD (**c**) values for each crown material–finish line group. Circles denote outliers with the test sample numbers displayed.

**Table 1 polymers-13-04311-t001:** Mean ± SD of MG, IG, and AMD of PICN and LDS crowns (n = 10).

Material	MG (µm)	IG (µm)	AMD (µm)
PICN-S	46.10 + 25.95	187.17 ± 58.78	95.48 ± 25.80
PICN-C	25.48 + 16.99	169.79 ± 19.38	122.44 ± 22.21
LDS-S	54.91 ± 25.28	143.81 ± 26.88	87.83 ± 26.21
LDS-C	59.03 ± 21.79	179.58 ± 31.20	89.99 ± 26.57

Using two-way ANOVA, the difference in MG and AMD between PICN and LDS crowns was found to be significant (*p* < 0.05) (Table 2), although the non-parametric Mann–Whitney U test showed that the differences in IG between the two materials were not significant (*p =* 0.253). Regardless of material, the MG and AMD differences between shoulder and chamfer margins were insignificant, and so were the interactions between ‘material’ and ‘finish line’ using two-way ANOVA (*p* > 0.05) (Table 2). The differences between the two finish lines were again not significant for IG using the Mann–Whitney U test (*p =* 0.060).

**Table 2 polymers-13-04311-t002:** Two-way ANOVA for MG and AMD.

Variables of Interest	Df	Sum of Squares	Mean Square	F	Sig. (*p*)
MG					
Material	1	4485.97	4485.97	8.64	0.006
PICN					
LDS					
Finish Line	1	681.07	681.07	1.31	0.26
Shoulder					
Chamfer	1				
Material * Finish Line		1529.38	1529.38	2.95	0.095
AMD					
Material	1	4021.25	4021.25	6.3	0.017
PICN					
LDS					
Finish Line	1	2119.223	2119.223	3.32	
Shoulder					0.077
Chamfer					
Material * Finish Line	1	1538.98	1538.98	2.41	0.129

The box plots (Figure 8) show the distribution of the MG, IG, and AMD values for the four material–finish line groups. The differences between the material–finish line groups for both MG and AMD were evaluated further by Bonferroni post-hoc analyses. PICN-C was not significantly different from PICN-S, but there were statistically significant differences between PICN-C and LDS-S (*p* = 0.039) and LDS-C (*p* = 0.013). As for AMD, the results were similar to MG. PICN-S and PICN-C showed insignificant differences, and so did LDS-S and LDS-C; however, PICN-C was significantly different from LDS-S (*p* = 0.025) and LDS-C (*p* = 0.041).

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
