# Peer review of "Effect of Finish Line Design on the Fit Accuracy of CAD/CAM Monolithic Polymer-Infiltrated Ceramic-Network Fixed Dental Prostheses: An In Vitro Study"

_polymers, 2021, doi:10.3390/polym13244311_

Round 1

Reviewer 1 Report

Dear Editor,

Regarding the submitted manuscript “  Effect of finish line design on the fit accuracy of CAD/CAM  monolithic polymer-infiltrated ceramic-network fixed dental prostheses” the presented study is intended to be an in vitro study to assess the effect of finish line design in the fit accuracy.

Overall appreciation

I think the authors should address some  corrections/clarifications and resubmit:

1-Abstract and title – It should be clearly stated in the title that it was an in vitro study since only in the end of the introduction that it is stated by the first time the type of study. Additionally, abbreviations should not be used in the abstract.  

2-The second null hypothesis should be rephrased from “…would be differences…” to “would be no differences”

3-Was the sample size adequate? – I cannot find any sample size calculation or a power analysis.  To address the sample size the authors, need to state what would be the mean differences to address with the proposed power.

5-In the material and methods section the authors should state:

  1. Why did the authors perform complete contour wax-ups instead of digital wax-ups?
  2. The randomization method used needs to be stated
  3. When stating intra-correlation (line 212 and 216) the authors need to indicate the 95% confidence intervals
  4. The statistics needs some professional help to be clearer in the description and some affirmations corrected. For example, what do the authors intend to say with “…through five statistics….” Line 240
  5. Tables and figures address the same results and should be reformulated

6-Discussion:

  1. The authors should reformulate the discussion to remove repeating results that were already reported before.
  2. Some minor misspellings should be corrected.
    • For example reformulate to partially reject instead of “Owing to the significant differences found in the MG and AMD values between the two material groups, this aspect of the first null hypothesis was rejected. However, in  terms of IG comparison for the two materials, the results affirm this part of the first null  ”
    • Replace CAD/CAD to CAD/CAM
    • Replace evalaution to evaluation
    • Replace crowsn to crowns
  3. The external validity should be better clarified since being an in vitro study , extrapolation should be performed with caution since biological behavior is different.

Based on the manuscript analysis I believe that the manuscript  should be considered for publication after reformulation.

Author Response

Dear Editor,

Regarding the submitted manuscript “ Effect of finish line design on the fit accuracy of CAD/CAM  monolithic polymer-infiltrated ceramic-network fixed dental prostheses” the presented study is intended to be an in vitro study to assess the effect of finish line design in the fit accuracy.

Overall appreciation

I think the authors should address some corrections/clarifications and resubmit:

1-Abstract and title – It should be clearly stated in the title that it was an in vitro study since only in the end of the introduction that it is stated by the first time the type of study. Additionally, abbreviations should not be used in the abstract.  

Response: Thank you for the comment. The required changes have been made according to the recommendation. Title and abstract have been modified to highlight the in-vitro aspect of the investigation. The abstract has also been altered to remove abbreviations wherever possible.

2-The second null hypothesis should be rephrased from “…would be differences…” to “would be no differences”

Response: The error has been corrected as per the advice (Introduction- last paragraph).

3-Was the sample size adequate? – I cannot find any sample size calculation or a power analysis.  To address the sample size the authors, need to state what would be the mean differences to address with the proposed power.

Response: The sample size calculation has been provided in the ‘Materials and method’ section (2.2. Preparation of the Crown Specimens, lines 134-139)

5-In the material and methods section the authors should state:

  1. Why did the authors perform complete contour wax-ups instead of digital wax-ups?

Response: Thank you for the observation. Yes, a digital anatomic crown wax-up could have also been used to fabricate standardized crowns on the dies, but however physical wax-ups were done and digitized (based on the methodology in reference no. 15)

  1. The randomization method used needs to be stated

Response: The randomization details have been included as advised (line 108).

  1. When stating intra-correlation (line 212 and 216) the authors need to indicate the 95% confidence intervals

Response: The reviewer’s comments have been taken into account and the concern addressed (Materials and Method- 2nd last paragraph).

  1. The statistics needs some professional help to be clearer in the description and some affirmations corrected. For example, what do the authors intend to say with “…through five statistics….” Line 240

Response: Professional assistance was sought again to confirm and to make the statistical explanation clearer, and the highlighted details corrected as advised.

  1. Tables and figures address the same results and should be reformulated

Response: Thank you very much for your valuable comment. While it is true that both the tables and the figures do provide some common information on the marginal gap, internal gap and AMD of the different crown-material finish line groups, however, in Table 1, the mean and standard deviations are provided, while the figure 2 show the spread of values (along with the median) for each of the outcome parameters. This has been clarified in the legend of Figure 2. So, the authors believe that both aspects of the information on MG, IG and AMD are equally important, and would like to retain these, if possible. Apart from this table and figure, there is no other overlap of information to the authors’ knowledge.  

6-Discussion:

  1. The authors should reformulate the discussion to remove repeating results that were already reported before.

Response: Thank you for your valuable comment. In the Discussion section paragraphs 2,3,4 and 5, the mean and SD values of MG (for PICN and LDS), IG and AMD are mentioned to objectively compare with the relevant studies in literature and note the similarities or provide reasons for the differences, if any. In Discussion paragraph 6 (lines 309-311) however, repeated results have been removed and discussion reorganized, as advised by the reviewer.

Other than these mentions, the authors have taken care to ensure that the results are not repeated in the discussion section, as advised.   

  1. Some minor misspellings should be corrected.
    • For example reformulate to partially reject instead of “Owing to the significant differences found in the MG and AMD values between the two material groups, this aspect of the first null hypothesis was rejected. However, in  terms of IG comparison for the two materials, the results affirm this part of the first null  ”

Response: The changes have been made in the discussion section (1st paragraph) as recommended.

    • Replace CAD/CAD to CAD/CAM

Response: The error has been corrected (Discussion section- paragraph 2, line 9).

    • Replace evalaution to evaluation

Response: This has been done (Discussion section- paragraph 2, line 10).

    • Replace crowsn to crowns

Response: The spelling error has been corrected ((Discussion section- paragraph 5, line 2).

  1. The external validity should be better clarified since being an in vitro study , extrapolation should be performed with caution since biological behavior is different.

Response: The authors totally agree with this comment and have added the necessary statement to highlight the clinical implications of this study (Discussion section- last paragraph).

Based on the manuscript analysis I believe that the manuscript should be considered for publication after reformulation.

Response: Thank you very much for your kind recommendation.

Reviewer 2 Report

The paper aims to evaluate the different fit of CAD/CAM fixed dental prostheses based on the combination of materials(Polymer infiltrated ceramic network vs Lithium disilicate) and finishing line design(chamfer vs shoulder).
The article is well structured and clear. 
Its relevance is due to the knowledge it adds regarding the finish line repercussion on the fit of Polimer infiltrated ceramic network (PICN) materials. The theme of finishing lines is to be shed light upon as it touches a less investigated variable, as the authors highlight by disagreeing in the conclusions with the scarce evidence nowadays present in regard to the new material tested.
Although there are already several articles that investigate the differences regarding the two materials, this article adds information extrapolated using new investigation methods such as a new model of an optical scanner since data obtained with such tools allowed to reliably investigate the evidence, these analysis tools are a “plus” which may be appropriate to conduct similar investigations in future research.
No self-citations or an abnormal number of citations were identified. No inappropriate references were cited.  The relevant references for contextualizing and basing the research are mostly within 5 years, except for citations concerning materials investigated in the past, on which research has slowed down, or used as a reference to compare the properties of new materials. In this case, older research is an aspect on which methodology relies upon. So it’s appropriate for authors to cite the older paper in this context.
The article shows no evident weakness, as is based on previously accepted methods in the literature for testing the proposed hypothesis. No methodological inaccuracies nor missing controls were observed. 
Clear information is given by the author about the methods used to acquire measures from the samples, such as clear identification on where to conduct measurements on the samples and the way that measurement should be conducted in particular shapes such as under contoured and over contoured crowns. 
Equipment used and sampling methods are described in detail so others can follow the same steps. 
Enough data are acquired to make the conclusions reliable.
Figures attached to the papers are helpful to the reader to understand at glance concepts hard to clarify solely with text, such as the positions where the measures were taken and the custom tool configuration used for analyzing the samples. 
Tables properly show the data acquired clearly, with good readability 
Conclusions are consistent with data evidence and arguments presented. They address the main questions posed. 
The research presents control experiments, repeated analyses, repeated experiments. Enough data are acquired to make the conclusions reliable. Sufficient data points do derive statistics with sufficient power to support the conclusions. No ethics statements were needed considering this is an in vitro study.

Author Response

Comments and Suggestions for Authors

The paper aims to evaluate the different fit of CAD/CAM fixed dental prostheses based on the combination of materials(Polymer infiltrated ceramic network vs Lithium disilicate) and finishing line design(chamfer vs shoulder).
The article is well structured and clear. 

Response: Thank you for your positive assessment.

Its relevance is due to the knowledge it adds regarding the finish line repercussion on the fit of Polimer infiltrated ceramic network (PICN) materials. The theme of finishing lines is to be shed light upon as it touches a less investigated variable, as the authors highlight by disagreeing in the conclusions with the scarce evidence nowadays present in regard to the new material tested.

Response: Yes, we concur with the reviewer.

Although there are already several articles that investigate the differences regarding the two materials, this article adds information extrapolated using new investigation methods such as a new model of an optical scanner since data obtained with such tools allowed to reliably investigate the evidence, these analysis tools are a “plus” which may be appropriate to conduct similar investigations in future research.

Response: The authors agree with the observations of the reviewer.

No self-citations or an abnormal number of citations were identified. No inappropriate references were cited.  The relevant references for contextualizing and basing the research are mostly within 5 years, except for citations concerning materials investigated in the past, on which research has slowed down, or used as a reference to compare the properties of new materials. In this case, older research is an aspect on which methodology relies upon. So it’s appropriate for authors to cite the older paper in this context.
The article shows no evident weakness, as is based on previously accepted methods in the literature for testing the proposed hypothesis. No methodological inaccuracies nor missing controls were observed. 
Clear information is given by the author about the methods used to acquire measures from the samples, such as clear identification on where to conduct measurements on the samples and the way that measurement should be conducted in particular shapes such as under contoured and over contoured crowns.

Response: Thank you for the constructive opinion on the methodology.

Equipment used and sampling methods are described in detail so others can follow the same steps. 
Enough data are acquired to make the conclusions reliable.
Figures attached to the papers are helpful to the reader to understand at glance concepts hard to clarify solely with text, such as the positions where the measures were taken and the custom tool configuration used for analyzing the samples. 
Tables properly show the data acquired clearly, with good readability 
Conclusions are consistent with data evidence and arguments presented. They address the main questions posed. 
The research presents control experiments, repeated analyses, repeated experiments. Enough data are acquired to make the conclusions reliable. Sufficient data points do derive statistics with sufficient power to support the conclusions. No ethics statements were needed considering this is an in vitro study.

Response: Thank you for the overall assessment.
